# Higher-Order Causal Message Passing for Experimentation with Complex Interference

**Mohsen Bayati**[1]    **Yuwei Luo**[1]    **William Overman**[1]    **Sadegh Shirani**[1]    **Ruoxuan Xiong**[2]

[1] Stanford Graduate School of Business    [2] Emory University
{bayati, yuweiluo, wpo, sshirani}@stanford.edu,  ruoxuan.xiong@emory.edu

## Abstract

Accurate estimation of treatment effects is essential for decision-making across various scientific fields. This task, however, becomes challenging in areas like social sciences and online marketplaces, where treating one experimental unit can influence outcomes for others through direct or indirect interactions. Such interference can lead to biased treatment effect estimates, particularly when the structure of these interactions is unknown. We address this challenge by introducing a new class of estimators based on causal message-passing, specifically designed for settings with pervasive, unknown interference. Our estimator draws on information from the sample mean and variance of unit outcomes and treatments over time, enabling efficient use of observed data to estimate the evolution of the system state. Concretely, we construct non-linear features from the moments of unit outcomes and treatments and then learn a function that maps these features to future mean and variance of unit outcomes. This allows for the estimation of the treatment effect over time. Extensive simulations across multiple domains, using synthetic and real network data, demonstrate the efficacy of our approach in estimating total treatment effect dynamics, even in cases where interference exhibits non-monotonic behavior in the probability of treatment.

## 1   Introduction

Randomized experiments are widely recognized as a reliable method in data-driven decision-making for determining the causal effects of new interventions, such as medical treatments or upgrades of market products. The conventional approach involves administering the new treatment to a randomly selected subset of the observation units (e.g., patients, products, or geographical areas), referred to as the *treatment* group, and comparing their outcomes with those units who received no treatment, the *control* group. However, the validity of these methods substantially relies on the assumption that treating a group of units does not interfere with the outcomes of the control units, known as the Stable Unit Treatment Value Assumption (SUTVA) [Cox, 1958, Rubin, 1978, Manski, 1990, Imbens and Rubin, 2015, Sussman and Airoldi, 2017].

In many social science and online marketplace scenarios, treating one unit impacts not only its outcome but also the outcomes of units that directly or indirectly interact with the treated unit [Bond et al., 2012, Blake and Coey, 2014, Holtz et al., 2020, Johari et al., 2022, Bright et al., 2022]. This *interference* of treatments and outcomes makes estimating the causal effect of the treatment particularly challenging. Considering the network of interactions, when a unit is treated, its interactions with neighboring units lead to subsequent changes in their outcomes. These interactions continue over the experimental time horizon and may display complex behaviors. For example, as the treatment is expanded to a larger population, the interference effect may intensify or diminish. This necessitates efficient data usage and robust estimators to capture and adapt to such intricacies.

Given the complexity of analyzing interference phenomena, research on network interference often relies on a series of simplifying assumptions. One common assumption is to ignore variations over time and assume outcomes are observed at equilibrium, which discards valuable information before the system reaches equilibrium. To reduce the complexity of the analysis, further assumptions are imposed on the nature and level of interference [Choi, 2017, Cortez et al., 2022, Li and Wager, 2022a], such as the neighborhood interference assumption or assumptions on the maximum degree of the network. Additionally, a frequently made assumption to help estimate treatment effects is that the interference network is observed [Chen et al., 2024, Agarwal et al., 2022, Jia et al., 2024], which is impractical in some settings, such as under pervasive interference. For example, in large-scale online platforms, units may interact through competing platforms, making it difficult to account for all sources of interference. Our aim in this paper is to relax these assumptions.

The impact of network interference can be intricate, particularly when considering interactions among units over time. For example, applying the treatment to one unit can have *spillover effects* on some control units, or one unit's outcome can directly exert *peer effects* on other units' outcomes. Simultaneously, treatments with long-lasting effects can have *carryover effects* to future time periods, and units' outcomes can be serially correlated or have *autocorrelation* over time. Consequently, whenever SUTVA fails to hold, the number of potential outcomes grows exponentially with the population size and the time horizon of the experiment. This renders the estimation of causal effects under general interference structures impossible due to non-identifiability challenges [Manski, 2013, Aronow and Samii, 2017, Basse and Airoldi, 2018, Karwa and Airoldi, 2018, Forastiere et al., 2022].

Recently, Shirani and Bayati [2024] introduced a new framework called *Causal Message-Passing* (CMP) to address the challenge of causal effect estimation under unobserved pervasive interference. Their methodology relies on observing outcomes over time and is rooted in statistical physics [Mezard et al., 1986, Mezard and Montanari, 2009] and approximate message passing (AMP) [Donoho et al., 2009, Bayati and Montanari, 2011] from high dimensional statistics. Instead of investigating the complex relationships among units, which requires knowledge of the network, CMP focuses on the dynamics of one-dimensional quantities, such as the sample mean and sample variance of units' outcomes over time. These one-dimensional equations, also known as *state evolution* equations, can help track how the administered intervention propagates through the network of units over time, which enables the estimation of counterfactual scenarios. However, it remains underexplored how to use state evolution to estimate causal effects.

In this work, we propose to utilize machine learning to learn a mapping that updates key parameters of the distribution of outcomes over time for causal effect estimation. This is achieved by introducing a set of non-linear feature functions that act on the observed outcomes, creating a "basis" for the learning task. By training a properly designed machine learning model on this extracted basis, we estimate the Total Treatment Effect (TTE), also known as the Global Treatment Effect (GTE) or Global Average Treatment Effect (GATE), which measures the causal effect of altering the treatment scenario from treating no one to treating everyone. The result is a family of estimators that allow one to extract more information from the experimental data, thereby ensuring efficient use of the data.

To be more specific, this work builds on the foundation established by Shirani and Bayati [2024], extending their method in two directions by introducing Higher-Order Causal Message Passing (HO-CMP) algorithms. First, HO-CMP incorporates higher-order moments of unit outcomes, unlike Shirani and Bayati [2024]'s approach, which only employs the first moments for estimation. Second, while Shirani and Bayati [2024] focus solely on two-stage experiments with two different probabilities of treatment, our work leverages the additional data provided by having more than two experimental stages with multiple probabilities of treatment. Thus, our work aligns with the common practice in the tech industry of rolling out treatments through a sequence of experiments [Kohavi et al., 2020].

We then validate the performance of HO-CMP by simulating multiple experimental settings, encompassing both linear and non-linear outcome specifications and various types of interference, such as synthetic random geometric networks and real-world networks. Specifically, we introduce a *Non-LinearInMeans* outcome specification, where the spillover effect is non-monotone in the fraction of treated neighbors; as an example of a complex treatment effect structure, we demonstrate how HO-CMP successfully estimates the total treatment effect by effectively utilizing higher-order moments of unit outcomes.

Simulating the experiments also allows us to calculate the ground truth value of the TTE, which remains unknown in real experiments, enabling us to compare the performance of HO-CMP to

the ground truth TTE. Additionally, we benchmark HO-CMP against standard approaches such as difference-in-means and Horvitz-Thompson estimators, a recent technique of Cortez et al. [2022], and a first-order CMP estimation, like the one by Shirani and Bayati [2024]. We emphasize that a large body of recent estimators, e.g., Jia et al. [2024], requires knowledge of the interference network and is not applicable in our setting. The results showcase HO-CMP outperforming the benchmarks in estimating the TTE over time and its flexibility to cover different outcome specifications and interference structures.

**Related causal inference literature.** The primary objective of research on causal inference in the context of network interference is to estimate causal effects while relaxing SUTVA. For this purpose, various assumptions and methods have been proposed. We briefly discuss the predominant ones.

A common approach to relax SUTVA is partial interference. Under this assumption, units are divided into disjoint clusters and interference is assumed only within the same cluster [Sobel, 2006, Rosenbaum, 2007, Hudgens and Halloran, 2012, Tchetgen and VanderWeele, 2012, Liu and Hudgens, 2014, Kang and Imbens, 2016, Viviano, 2020b, Bhattacharya et al., 2020, Qu et al., 2021, Auerbach and Tabord-Meehan, 2021, Candogan et al., 2023, Ugander and Yin, 2023]. When interference extends across clusters, standard estimators become biased. To address this, Eckles et al. [2016] propose a cluster-randomized approach that randomizes treatment assignment across clusters, reducing bias. However, it requires knowledge of the clusters.

The other assumption to replace SUTVA is the Neighborhood Interference Assumption (NIA). NIA states that outcomes are only influenced by the treatments of neighboring units in the network. This assumption is commonly imposed in the literature that relaxes the SUTVA [Sussman and Airoldi, 2017]. Some recent studies combine the NIA with the availability of either a fully or partially observed interference structure [Leung, 2020, Viviano, 2020a, Agarwal et al., 2022, Belloni et al., 2022, Li and Wager, 2022b]. Without prior knowledge of the interference structure, Cortez et al. [2022] consider low-degree polynomial interactions among units in the network. Leung [2022] also introduces a weaker version of the NIA, where the interference between two units located far away from each other is allowed to be nonzero, but negligible.

Another approach is to facilitate the estimation of causal effects by setting restrictions on the network structure [Chin, 2018, Jagadeesan et al., 2020, Wang et al., 2020, Li and Wager, 2022a, Agarwal et al., 2022, Jagadeesan et al., 2020, Leung, 2022]. These restrictions include bounding the largest node degree of the interference graph, limiting the degree of the dependency graph, observing specific patterns in the network, locally constrained interference structures, and restricting the topology of the interference network.

Driven by applications in marketplace platforms and two-sided marketplaces, several recent works have examined specific interference patterns [Holtz et al., 2020, Wager and Xu, 2021, Munro et al., 2021, Johari et al., 2022, Harshaw et al., 2022, Farias et al., 2022, Bright et al., 2022, Farias et al., 2023]. For example, Farias et al. [2022] study experiments in Markovian systems where interference effects propagate through constraints like limited inventory.

From another perspective, most of the existing literature on network interference focuses on the case of single-time point observation [Hudgens and Halloran, 2012, Aronow and Samii, 2017, Basse et al., 2019, Jackson et al., 2020, Sävje et al., 2021]. These studies have provided insightful results on spatial interference effects, but they often overlook temporal variations of the treatment effect. Recently, there has been a shift to consider settings with multiple-time observations [Li and Wager, 2022a, Boyarsky et al., 2023]. However, the problem of considering the dynamics of units' outcomes remains understudied [Arkhangelsky and Imbens, 2023].

## 2   Setup and Foundation

Consider a system of $N$ units indexed by $i \in [N] := \{1, \cdots, N\}$ subject to a randomized experiment. The units are observed over a time horizon of $T + 1$ periods and for each $t \in \{0, 1, \ldots, T\}$, we let $W_t^i$ denote the treatment status of unit $i$ during time period $t$. For simplicity, we consider a Bernoulli randomized design such that $W_t^i \sim \text{Bernoulli}(\pi_t)$. That is, at time $t$ unit $i$ receives the *treatment* with a probability of $\pi_t$, corresponding to $W_t^i = 1$. Otherwise, unit $i$ belongs to the *control* group and $W_t^i = 0$. In this context, we collectively define $\boldsymbol{\pi} = (\pi_0, \pi_1, \ldots, \pi_T)$ as the *experimental design*. Then, following the potential outcome framework [Imbens and Rubin, 2015], let $Y_t^i(\boldsymbol{W})$ represent

the potential outcome of unit $i$ at time $t$, where $\boldsymbol{W}$ denotes the entire treatment allocation matrix, with $W_t^i$ as the entry in row $t$ and column $i$.

Administering the treatment of unit $i$ at time $t$ according to $w_t^i$ (as one realization of the random variable $W_t^i$), we use $\boldsymbol{w}$ (as one realization of $\boldsymbol{W}$) to show the matrix that captures the treatments of all units throughout the experiment; accordingly, we let $y_t^i = Y_t^i(\boldsymbol{W} = \boldsymbol{w})$ be the observed outcome of unit $i$ at time $t$ under the treatment assignment $\boldsymbol{w}$:

$$
\boldsymbol{w} = \begin{bmatrix} w_0^1 & w_0^2 & \dots & w_0^N \\ w_1^1 & w_1^2 & \dots & w_1^N \\ \vdots & \vdots & \ddots & \vdots \\ w_T^1 & w_T^2 & \dots & w_T^N \end{bmatrix}, \qquad \boldsymbol{y} = \begin{bmatrix} y_0^1 & y_0^2 & \dots & y_0^N \\ y_1^1 & y_1^2 & \dots & y_1^N \\ \vdots & \vdots & \ddots & \vdots \\ y_T^1 & y_T^2 & \dots & y_T^N \end{bmatrix}.
$$

Observing $(\boldsymbol{w}, \boldsymbol{y})$, we are interested in estimating the TTE of the intervention, defined as below:

$$
\text{TTE}_t = \lim_{N \to \infty} \frac{1}{N} \sum_{i=1}^{N} \left[ Y_t^i(\boldsymbol{1}) - Y_t^i(\boldsymbol{0}) \right], \quad t = 0, 1, \dots, T, \tag{1}
$$

where $\boldsymbol{1}$ and $\boldsymbol{0}$ are matrices of all 1 and all 0 of appropriate dimensions (in this case, $T+1$ by $N$). Intuitively, the TTE measures the average effect of changing the treatment for the entire population. This is a common estimand in the network interference literature and provides important insights into the efficacy of the treatment for decision-makers [Jia et al., 2024, Chen et al., 2024, Viviano et al., 2023, Yu et al., 2022, Cortez et al., 2022].

Deriving a practical and efficient estimator for the TTE is challenging due to the fact that we can observe the population only under one treatment scenario [Holland, 1986]. Indeed, in Eq. (1), we can observe at most one of $Y_t^i(\boldsymbol{1})$ or $Y_t^i(\boldsymbol{0})$, and often, neither.[1] In the following sections, we address this challenge by proposing a new class of estimators grounded in the CMP framework. These estimators rely on the efficient use of experimental data, $\boldsymbol{y}$ and $\boldsymbol{w}$, yielding accurate causal estimation under unknown network interference.

## 2.1 Potential outcome specification and state evolution of the experiment

In this section we provide a summary of the outcome specification and results of Shirani and Bayati [2024] that we utilize in the remaining. For $t = 0, 1, \dots, T - 1$, we let $g_t : \mathbb{R} \times \mathbb{R}^{T+1} \mapsto \mathbb{R}$ be an unknown measurable function. We also use $\vec{W}^i = \left( W_0^i, \dots, W_T^i \right)^\top$ to denote the treatment assignment of unit $i$ during the experiment. Accordingly, the treatment allocation matrix $\boldsymbol{W}$ is a $T+1$ by $N$ matrix with columns equal to $\vec{W}^i$. Given potential outcomes $Y_t^j(\boldsymbol{W})$ at time $t$ and $j \in [N]$, their outcomes in time period $t + 1$ are specified by

$$
Y_{t+1}^i(\boldsymbol{W}) = \sum_{j=1}^{N} \mathrm{G}^{ij} g_t \left( Y_t^j(\boldsymbol{W}), \vec{W}^j \right) + \epsilon_t^i, \qquad t = 0, 1, \dots, T - 1, \tag{2}
$$

where $\mathrm{G}^{ij}$ quantifies the impact of unit $j$ on unit $i$ at time $t$ and $\epsilon_t^i$ is a zero-mean Gaussian noise with a variance of $\sigma_e^2$, accounting for measurement errors. In addition, we let $\mathbf{G} = [\mathrm{G}^{ij}]_{i,j \in [N]}$ and refer to it as the *interference matrix*. Then, according to Eq. (2), the function $g_t$ captures the impact of past outcomes and treatment assignments of other units on the current outcome of unit $i$.

Now, fixing $t$, we define

$$
\nu_t(\boldsymbol{W}) := \lim_{N \to \infty} \frac{1}{N} \sum_{i=1}^{N} Y_t^i(\boldsymbol{W}), \qquad \rho_t(\boldsymbol{W})^2 := \lim_{N \to \infty} \frac{1}{N} \sum_{i=1}^{N} Y_t^i(\boldsymbol{W})^2 - \nu_t(\boldsymbol{W})^2. \tag{3}
$$

Then, as shown by Shirani and Bayati [2024], whenever the elements of the interference matrix $\mathrm{G}^{ij}$ are i.i.d. Gaussian random variables with mean $\mu/N$ and variance $\sigma^2/N$, under mild moment conditions on initial values $Y_0^i$, we have

$$
\begin{aligned}
\nu_{t+1}(\boldsymbol{W}) &\stackrel{\text{a.s.}}{=} \mu \mathbb{E} \left[ g_t \big( \nu_t(\boldsymbol{W}) + \rho_t(\boldsymbol{W}) Z_t, \vec{W} \big) \right], \\
\rho_{t+1}(\boldsymbol{W})^2 &\stackrel{\text{a.s.}}{=} \sigma^2 \mathbb{E} \left[ g_t \big( \nu_t(\boldsymbol{W}) + \rho_t(\boldsymbol{W}) Z_t, \vec{W} \big)^2 \right] + \sigma_e^2,
\end{aligned} \tag{4}
$$

---

[1] We can only observe $\boldsymbol{y}$ for one of exponentially many realizations of $\boldsymbol{w}$.

where $Z_t \sim \mathcal{N}(0,1)$ is independent from $\vec{W} \sim \text{Bernoulli}(\boldsymbol{\pi})$ (that is, $W_t \sim \text{Bernoulli}(\pi_t)$ and $\vec{W} = (W_0, W_1, \ldots, W_T)^\top$) and the equalities hold almost surely. We note that the theory behind this result is rooted in the AMP literature, going back to Bolthausen [2014], Bayati and Montanari [2011]. However, as Shirani and Bayati [2024] note, there is a major distinction between the AMP literature and the above setting: in the AMP literature, the matrix $\mathbf{G}$ is observed, and the aim is to construct proper functions $g_t$ for a completely different objective, which is studying the high-dimensional asymptotics of first-order algorithms. However, in the current context, the matrix $\mathbf{G}$ and functions $g_t$ are *unknown* and the goal is to estimate them.

Considering Eq. (3), the equations in (4) determine the dynamics of the sample mean and sample variance of unit outcomes over time in large sample asymptotics, and are denoted by the State Evolution (SE) equations of the experiment [Shirani and Bayati, 2024]. In the next section, we present an efficient algorithm to learn the state evolution dynamics outlined in Eq. (4). This method enables us to accurately estimate the TTE defined in Eq. (1) and its corresponding confidence interval.

## 3  Algorithm

In this section, we introduce *Higher-order Causal Message-passing* (HO-CMP) for estimating the TTE over the entire time horizon of the experiment. Briefly speaking, HO-CMP directly estimates the update function in the state evolution equations (4), thereby estimating counterfactual quantities while accounting for the impact of unknown network interference. To this end, by Eqs. (1) and (3), we rewrite the TTE as the difference of the sample means in the large limits:

$$\text{TTE}_t = \nu_t(\mathbf{1}) - \nu_t(\mathbf{0}).$$

That means the problem of estimating the TTE is equivalent to estimating $\nu_t(\mathbf{1})$ and $\nu_t(\mathbf{0})$ using the observed data, denoted by $(\boldsymbol{w}, \boldsymbol{y})$. On the other hand, considering the state evolution equations in (4), the *system state* at time $t+1$, denoted by $(\nu_{t+1}(\boldsymbol{w}), \rho_{t+1}(\boldsymbol{w})^2)$, is a (nonlinear) function of the system state distribution at time $t$, characterized by $(\nu_t(\boldsymbol{w}), \rho_t(\boldsymbol{w})^2)$ and $\vec{W}$, encompassing the sample mean and variance of observed outcomes as well as the design of the experiment. However, because the exact functional form and parameters of equations in (4) are unknown, one cannot directly apply the SE to track the evolution of states. Therefore, we propose to estimate the unknown update functions in SE equations, utilizing the observed data $(\boldsymbol{w}, \boldsymbol{y})$. For this purpose, we fix the treatment assignment matrix $\boldsymbol{w}$ and define

$$\hat{\nu}_t(\boldsymbol{w}) := \frac{1}{N} \sum_{i=1}^N y_t^i, \qquad \hat{\rho}_t(\boldsymbol{w})^2 := \frac{1}{N} \sum_{i=1}^N \left( y_t^i - \hat{\nu}_t(\boldsymbol{w}) \right)^2,$$

$$\bar{w}_t := \frac{1}{N} \sum_{i=1}^N w_t^i, \qquad \vec{\bar{w}} := (\bar{w}_0, \ldots, \bar{w}_T)^\top.$$

In addition, let $\vec{\phi} = (\phi_k)_{k \in [K]}$ be a prespecified vector of measurable feature functions of current estimates of the sample mean $\hat{\nu}_t(\boldsymbol{w})$, sample variance $\hat{\rho}_t(\boldsymbol{w})^2$, and the design $\boldsymbol{w}$. We define $\boldsymbol{x}_t$ to represent the *feature vector* as follows:

$$\boldsymbol{x}_t = \vec{\phi}\Big(\hat{\nu}_t(\boldsymbol{w}), \hat{\rho}_t(\boldsymbol{w}), \boldsymbol{w}\Big) := \Big[\phi_1\Big(\hat{\nu}_t(\boldsymbol{w}), \hat{\rho}_t(\boldsymbol{w}), \boldsymbol{w}\Big), \ldots, \phi_K\Big(\hat{\nu}_t(\boldsymbol{w}), \hat{\rho}_t(\boldsymbol{w}), \boldsymbol{w}\Big)\Big]^\top.$$

Then, we formally propose learning the mapping $f_{\boldsymbol{\theta}}(\cdot)$ defined by,

$$(\hat{\nu}_{t+1}(\boldsymbol{w}), \hat{\rho}_{t+1}(\boldsymbol{w})^2) = f_{\boldsymbol{\theta}}(\boldsymbol{x}_t) \tag{5}$$

We summarize the method in Algorithm 1. Note in our experiment design we begin with all units under control by setting $\pi_0 = 0$, meaning no units receive treatment in period 0. Additionally, to avoid non-identifiability issues, the experiment requires at least two stages, which corresponds to having at least two distinct values in the set $\{\pi_1, \ldots, \pi_T\}$.

The proposed HO-CMP method encompasses a rich family of estimators, offering flexibility through the selection of feature functions $\{\phi_k\}_{k \in [K]}$ and model $f_{\boldsymbol{\theta}}(\cdot)$. Specifically, incorporating proper feature (basis) functions, with examples shown in Table 1, facilitates the extraction of informative patterns for learning the unknown nonlinear dynamics of the system throughout the experiment. In

Table 1: Two examples of feature functions

| Algorithms | Feature functions $\{\phi_k(\hat{\nu}_t(\boldsymbol{w}), \hat{\rho}_t(\boldsymbol{w})^2, \boldsymbol{w})\}_{k\in[K]}$ | $f_{\boldsymbol{\theta}}(\cdot)$ |
|---|---|---|
| FO-CMP | $\{\hat{\nu}_t(\boldsymbol{w}), \bar{w}_{t+1}, \hat{\nu}_t(\boldsymbol{w}) \cdot \bar{w}_t\}$ | linear regression |
| HO-CMP | $\{\hat{\nu}_t(\boldsymbol{w}), \bar{w}_{t+1}, \hat{\nu}_t(\boldsymbol{w}) \cdot \bar{w}_t, \hat{\rho}_t(\boldsymbol{w})^2, \bar{w}_{t+1}^2\}$ | linear regression |

practice, one could choose these basis functions based on heuristics, domain knowledge, and prior information about the dynamics.

Specifically, in this paper, we consider the following estimators, as summarized in Table 1.

FO-CMP (First-Order Causal Message-Passing): This corresponds to the simple setting where $\nu_{t+1}(\boldsymbol{w})$ is assumed to be a function of the previous sample mean $\nu_t(\boldsymbol{w})$, the sample mean of the current treatment $\bar{w}_{t+1}$, and an additional term to model the interaction of the dynamics and previous treatments $\nu_t(\boldsymbol{w})\bar{w}_t$. Consequently, this model is irrelevant of the variance $\rho_{t+1}(\boldsymbol{w})^2$. This is true when $g_t$ takes a simple nonlinear form $g_t(y_t, \vec{w}) = \alpha y_t + \beta w_{t+1} + \gamma y_t w_t$. We remark that FO-CMP essentially uses the first state evolution equation in (4) and fails to extract informative signals from the second evolution equation.

HO-CMP (Higher-Order Causal Message-Passing): HO-CMP further introduces the second-order terms $(\bar{w}_{t+1})^2$ and $\hat{\rho}_t(\boldsymbol{w})^2$ to model the nonlinear treatment effects. It improves data efficiency by utilizing both state evolution equations. It also allows estimation of higher order terms in Taylor series of $g_t$.

While FO-CMP extends the estimation algorithm in Shirani and Bayati [2024] to accommodate experiments with more than two stages, HO-CMP introduces a new dimension to the estimation problem by incorporating second-order terms. This inclusion enhances data utilization, resulting in higher estimation efficiency in HO-CMP compared to FO-CMP.

---

**Algorithm 1:** Higer-Order Causal Message Passing (HO-CMP)

---

**Data:** Observed data $(\boldsymbol{w}, \boldsymbol{y})$, feature functions $\vec{\phi} = (\phi_k)_{k\in[K]}$, machine learning model $f_{\boldsymbol{\theta}}(\cdot)$
**Step 1: Data processing**
  **for** $t \leftarrow 0$ **to** $T$ **do**
   $\hat{\nu}_t(\boldsymbol{w}) \leftarrow \frac{1}{N}\sum_{i=1}^N y_t^i$,
   $\hat{\rho}_t(\boldsymbol{w})^2 \leftarrow \frac{1}{N}\sum_{i=1}^N (y_t^i - \hat{\nu}_t(\boldsymbol{w}))^2$,
   $\boldsymbol{x}_t \leftarrow \vec{\phi}\left(\hat{\nu}_t(\boldsymbol{w}), \hat{\rho}_t(\boldsymbol{w})^2, \boldsymbol{w}\right)$
  **end**
**Step 2: Model Estimation**
  Estimate $f_{\boldsymbol{\theta}}$ from data $\left\{\left(\boldsymbol{x}_t, \left(\hat{\nu}_{t+1}(\boldsymbol{w}), \hat{\rho}_{t+1}(\boldsymbol{w})^2\right)\right)\right\}_{t\in[T-1]}$, guided by (5).
**Step 3: Counterfactual Estimation**
  $\hat{\nu}_0(\boldsymbol{0}) \leftarrow \hat{\nu}_0(\boldsymbol{w}), \hat{\nu}_0(\boldsymbol{1}) \leftarrow \hat{\nu}_0(\boldsymbol{w}), \hat{\rho}_0(\boldsymbol{0})^2 \leftarrow \hat{\rho}_t(\boldsymbol{w})^2, \hat{\rho}_0(\boldsymbol{1})^2 \leftarrow \hat{\rho}_t(\boldsymbol{w})^2, \widehat{\text{TTE}}_0 \leftarrow 0$
  **for** $t \leftarrow 0$ **to** $T-1$ **do**
   Compute the features and predict the counterfactuals
   $\boldsymbol{x}_t(\boldsymbol{0}) \leftarrow \vec{\phi}\left(\hat{\nu}_t(\boldsymbol{0}), \hat{\rho}_t(\boldsymbol{0})^2, \boldsymbol{0}\right), \boldsymbol{x}_t(\boldsymbol{1}) \leftarrow \vec{\phi}\left(\hat{\nu}_t(\boldsymbol{1}), \hat{\rho}_t(\boldsymbol{1})^2, \boldsymbol{1}\right)$
   $\left(\hat{\nu}_{t+1}(\boldsymbol{0}), \hat{\rho}_{t+1}(\boldsymbol{0})^2\right) \leftarrow f_{\boldsymbol{\theta}}(\boldsymbol{x}_t(\boldsymbol{0})), \left(\hat{\nu}_{t+1}(\boldsymbol{1}), \hat{\rho}_{t+1}(\boldsymbol{1})^2\right) \leftarrow f_{\boldsymbol{\theta}}(\boldsymbol{x}_t(\boldsymbol{1}))$
   Estimate the TTE
   $\widehat{\text{TTE}}_{t+1} \leftarrow \hat{\nu}_{t+1}(\boldsymbol{1}) - \hat{\nu}_{t+1}(\boldsymbol{0})$
  **end**
**Result:** $\left\{\widehat{\text{TTE}}_t\right\}_{t\in[T]}$

---

# 4 Experiments

In this section, we use synthetic experiments under simulated and real-world network interference patterns, to compare the performance of FO-CMP and HO-CMP estimators, outlined in Table 1 and Algorithm 1, with several benchmarks. First, we introduce the experimental design, benchmark estimators, interference patterns, and outcome specifications.

**Experimental design.** We primarily focus on the staggered rollout design with $L$ distinct treated probabilities, denoted by $\pi^{(1)}, \cdots, \pi^{(L)}$, where $\pi^{(\ell)}$ increases monotonically with $\ell \in \{1, \ldots, L\}$. In the first $T^{(1)}$ periods, $\pi^{(1)} \times 100\%$ of units are in the treatment group. From $T^{(1)}$ to $T^{(2)}$ periods, $\pi^{(2)} \times 100\%$ of units are in the treatment group, and so forth. In the staggered rollout design, once a unit is allocated to treatment, it remains in the treatment group until the experiment concludes [Xiong et al., 2024]. In the appendix, we also consider the Bernoulli randomized design, where the treatment is re-randomized at every time period, allowing units to switch between the treatment and control groups throughout the experiment. We use two values of $T = 40, \ 200$ and set $L = 4$, with $(\pi^{(1)}, \pi^{(2)}, \pi^{(3)}, \pi^{(4)}) = (0.1, 0.2, 0.4, 0.5)$. In the appendix, we show the impact of increasing $L$ or the maximum treatment probability $\pi^{(L)}$.

**Benchmark estimators.** We first present two benchmark estimators commonly used for treatment effect estimation, both in settings with and without network interference. The final estimator is designed specifically for settings with unknown network interference [Cortez et al., 2022].

The first benchmark estimator is the standard difference-in-means (DM) estimator given by

$$\widehat{\text{TTE}}_t^{\text{dm}} = \frac{\sum_{j=1}^N y_t^j w_t^j}{\sum_{j=1}^N w_t^j} - \frac{\sum_{j=1}^N y_t^j (1 - w_t^j)}{\sum_{j=1}^N (1 - w_t^j)},$$

which is the difference in average outcomes between treated and control units at each time period $t$.

The second benchmark is the standard Horvitz and Thompson [1952] (HT) estimator given by

$$\widehat{\text{TTE}}_t^{\text{ht}} = \frac{1}{N} \sum_{j=1}^N \left[ \frac{y_t^j w_t^j}{\pi_t} - \frac{y_t^j (1 - w_t^j)}{1 - \pi_t} \right],$$

which weights observed outcomes by the inverse propensity score (i.e., $1/\pi_t$ or $1/(1 - \pi_t)$).

The third benchmark estimator is the polynomial interpolation estimator (PolyFit) introduced by Cortez et al. [2022]. PolyFit operates by obtaining estimates for the average of outcomes at equilibrium for $L$ treated probabilities $\pi^{(1)}, \cdots, \pi^{(L)}$, denoted by $\nu_{\text{equil}}(\pi^{(1)}), \ldots, \nu_{\text{equil}}(\pi^{(L)})$, then it utilizes Lagrange interpolation method and obtains a degree-$L$ polynomial approximation for the function $\nu_{\text{equil}} : [0, 1] \to \mathbb{R}$ which can be used to estimate the equilibrium values under global control and treatment, $\hat{\nu}_{\text{equil}}(0)$ and $\hat{\nu}_{\text{equil}}(1)$. Finally, TTE is estimated by

$$\widehat{\text{TTE}}_t^{\text{polyfit}} = \hat{\nu}_{\text{equil}}(1) - \hat{\nu}_{\text{equil}}(0).$$

On the one hand, PolyFit does not need any knowledge of the interference network; however, it comes at the expense of having to grapple with two challenges. First, it may incur a high variance as $L$ increases due to fitting a high-degree polynomial. The second challenge is that it needs accurate estimates for each $\nu_{\text{equil}}(\pi^{(\ell)})$, which requires treating $\pi^{(\ell)}$ fraction of units for a long enough number of periods so that the outcomes reach an equilibrium. This can be achieved if the staggered roll-out design is performed over a long enough horizon $T$ with each $T^{(\ell)}$ sufficiently large, and then estimating each $\nu_{\text{equil}}(\pi^{(\ell)})$ by sample average of outcomes at time $T^{(\ell)}$. However, when such a lengthy experiment is not feasible, the estimates for $\nu_{\text{equil}}(\pi^{(\ell)})$ will be less accurate.

**Interference networks.** We consider two networks (graphs). The first graph is a simulated random geometric graph model, studied by Leung [2022]. The second graph is a social network of Twitch users [Rozemberczki and Sarkar, 2021]. In either scenario, we denote the adjacency matrix of the graph by $E \in \{0, 1\}^{N \times N}$. For any $i$ and $j$, $E_{ij}$ equals 1 if $j$ is a neighbor of $i$ and 0 otherwise.

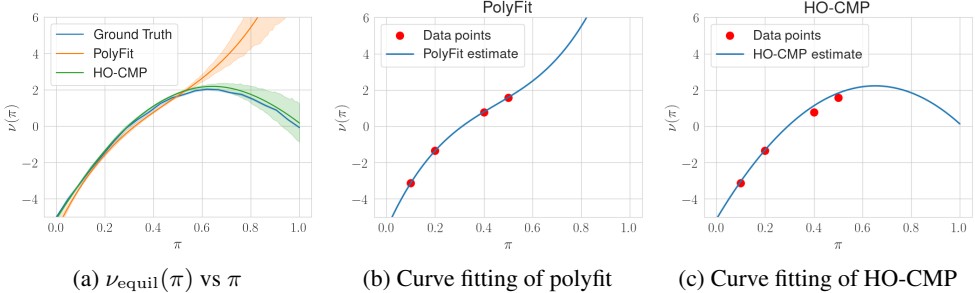

| (a) $\nu_{\text{equil}}(\pi)$ vs $\pi$ | (b) Curve fitting of polyfit | (c) Curve fitting of HO-CMP |

Figure 1: (a) $\nu_{\text{equil}}(\pi)$ with PolyFit and HO-CMP estimates across runs (Non-LinearInMeans). (b) and (c) show one sample estimates with observed data points.

**Outcome generating processes.** We consider two outcome specifications to model monotone and non-monotone interference patterns. Specifically, for both settings, we generate outcomes using the following specification:

$$Y_{t+1}^i = \alpha + \beta \frac{\sum_{j=1}^N E_{ij} Y_t^j}{\sum_{j=1}^N E_{ij}} + \delta \cdot g\left(\frac{\sum_{j=1}^N E_{ij} W_{t+1}^j}{\sum_{j=1}^N E_{ij}}\right) + \gamma W_{t+1}^i + \epsilon_{t+1}^i,$$

where in the first setting, $g(\cdot)$ is taken to be the identity function, i.e., $g(x) = x$ for any $x$. Therefore, $Y_{t+1}^i$ depends linearly on the fraction of treated neighbors, and we refer to this setting as the *LinearInMeans* outcome setting. This setting is widely studied in the causal inference literature [Cai et al., 2015, Eckles et al., 2016, Leung, 2022].

In the second setting, $g(\cdot)$ is specified by a periodic function, i.e., $g(x) = \sin(\pi x)$ for any $x$. Therefore, $Y_{t+1}^i$, on average, first increases and then decreases with the fraction of treated neighbors, as visualized by the Ground Truth curve in panel (a) of Figure 1. We refer to this setting as the *Non-LinearInMeans* outcome setting.

**Results.** We compare FO-CMP and HO-CMP with the three benchmarks for estimating the TTE across the aforementioned outcome specifications and interference networks for long ($T = 200$) and short ($T = 40$) horizons. In each scenario, we perform 100 simulations of the synthetic experiment. The resulting distributions of ground truth and estimated TTEs are shown in Figures 2-5. All experiments were conducted on a MacBook Air with an Apple M1 chip and 16 GB of memory, with each setting taking about 15 minutes for 100 iterations. The key takeaways are as follows.

First, the DM and HT estimators exhibit significant bias across all cases. This is intuitive, as they estimate the TTE without accounting for the network interference.

Second, in the *LinearInMeans* outcome setting, FO-CMP and HO-CMP achieve low estimation error and minimal bias. This holds for both long experiment durations ($T = 200$), where outcomes reach equilibrium, and short experiment durations ($T = 40$), where outcomes have not yet reached equilibrium, as shown in Figures 2 and 3, respectively.

Third, as expected, PolyFit's dependence on accurate estimates for each $\nu_{\text{equil}}(\pi^{(\ell)})$ requires a large $T$ to reduce estimation bias. This is evident when comparing Figures 2 and 3: with a smaller $T$, PolyFit shows bias. This is also demonstrated in panel (b) of Figure 1, where the red points—which represent sample averages of outcomes at $T^{(1)}, \ldots, T^{(4)}$—have not yet converged and are slightly lower than their ground truth (equilibrium) values. This causes PolyFit's estimation of $\hat{\nu}_{\text{equil}}(1)$ to be inaccurate, leading to a large bias. In contrast, HO-CMP, as shown in panel (c) of Figure 1, is immune to this problem as it is designed to work with off-equilibrium data. Even with a larger $T$, the degree-$L$ polynomial estimation costs PolyFit with higher variance than both FO-CMP and HO-CMP, as shown in Figure 2. Overall, this underscores the more efficient data utilization of FO-CMP and HO-CMP through their ability to leverage off-equilibrium data.

Fourth, in the *Non-LinearInMeans* outcome setting, HO-CMP achieves substantially lower estimation error compared to FO-CMP, as shown in Figures 4 and 5. This makes intuitive sense, as the higher-

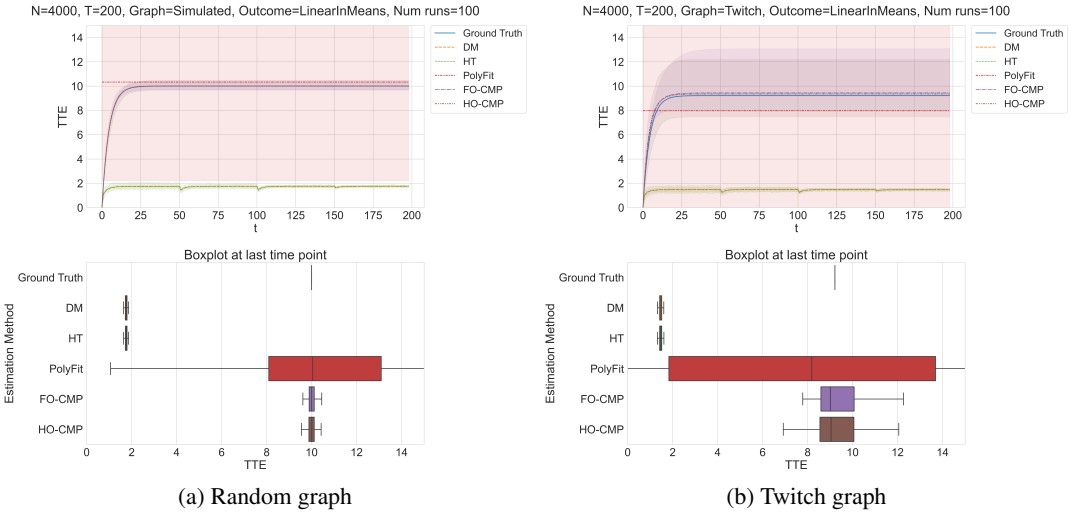

Figure 2: *LinearInMeans with $T = 200$. $L = 4$*, with $\left(\pi^{(1)}, \pi^{(2)}, \pi^{(3)}, \pi^{(4)}\right) = (0.1, 0.2, 0.4, 0.5)$ and $T^{(\ell)} = 50\ell$ for all $\ell \in \{1, 2, 3, 4\}$. Shaded regions show 95% percentile intervals of mean.

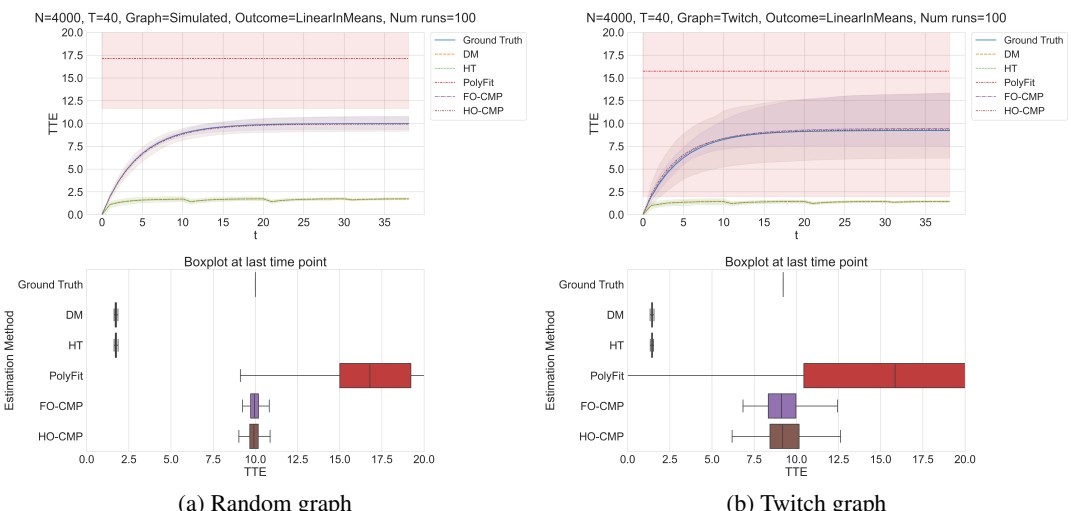

Figure 3: *LinearInMeans with $T = 40$. $L = 4$*, with $\left(\pi^{(1)}, \pi^{(2)}, \pi^{(3)}, \pi^{(4)}\right) = (0.1, 0.2, 0.4, 0.5)$ and $T^{(\ell)} = 10\ell$ for all $\ell \in \{1, 2, 3, 4\}$. Shaded regions show 95% percentile intervals of mean.

order terms in HO-CMP better capture the nonlinearity of $\nu_{\mathrm{equil}}(\pi)$ in $\pi$, while leveraging the additional data on sample variance dynamics over time, thereby enhancing the estimation accuracy.

Finally, the proposed estimation method demonstrates robustness across different experimental setups, including both *LinearInMeans* and *Non-LinearInMeans* outcome specifications. Additionally, robustness to graph structure—random versus Twitch graph—is evident from comparing the left and right plots in Figures 2-5. In Figure 6 of Appendix A, we also demonstrate the robustness of the proposed methods to various parameters: the number of treatment probabilities $L$, the maximum treatment probability $\pi^{(L)}$, and the choice of experimental design (staggered rollout versus Bernoulli randomization).

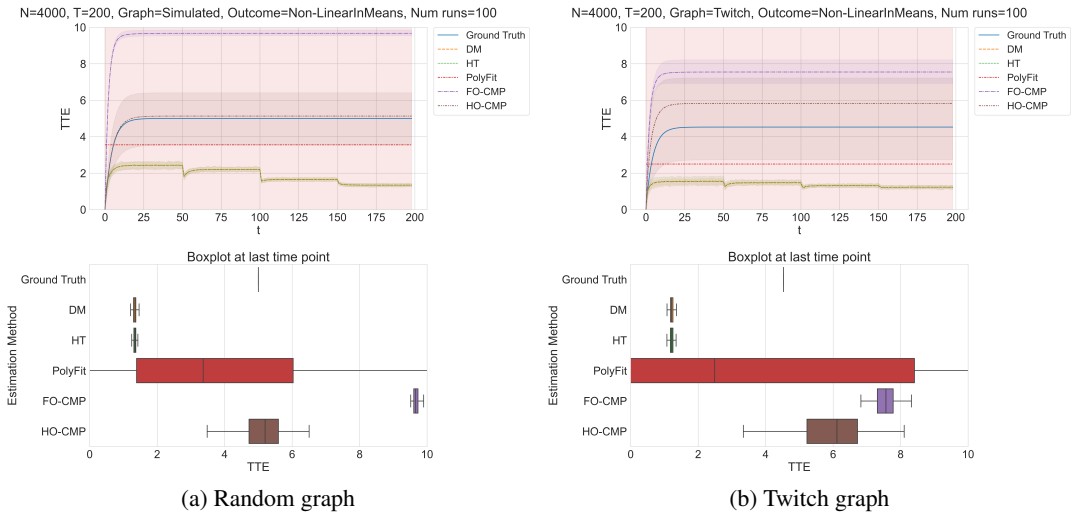

Figure 4: *Non-LinearInMeans with* $T = 200$. $L = 4$, with $\left(\pi^{(1)}, \pi^{(2)}, \pi^{(3)}, \pi^{(4)}\right) = (0.1, 0.2, 0.4, 0.5)$, and $T^{(\ell)} = 50\ell$ for all $\ell$. Shaded regions show 95% percentile intervals of mean.

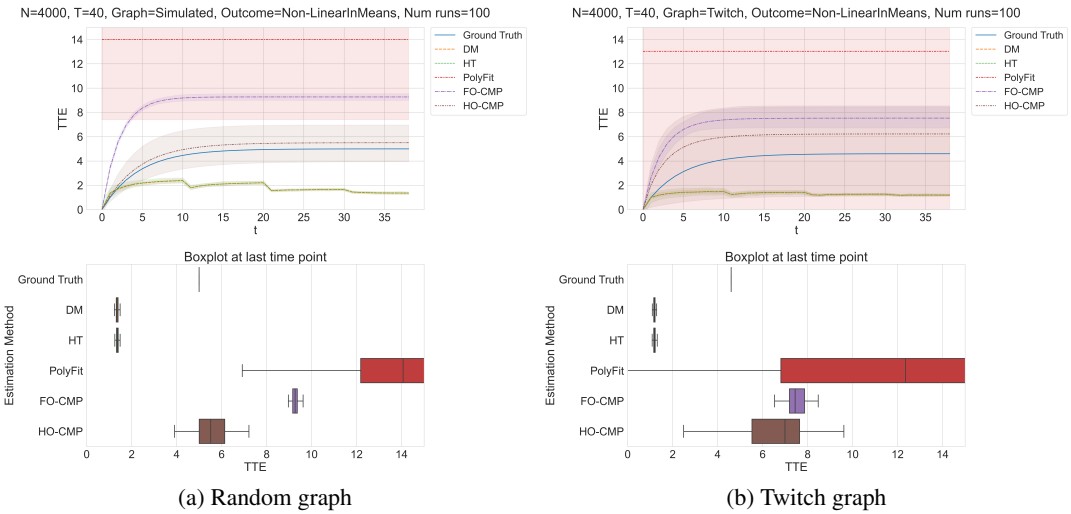

Figure 5: *Non-LinearInMeans with* $T = 40$. $L = 4$, with $\left(\pi^{(1)}, \pi^{(2)}, \pi^{(3)}, \pi^{(4)}\right) = (0.1, 0.2, 0.4, 0.5)$, and $T^{(\ell)} = 10\ell$ for all $\ell$. Shaded regions show 95% percentile intervals of mean.

## 5 Conclusion

Estimating causal effects under pervasive interference presents significant challenges [Sussman and Airoldi, 2017]. Building on the causal message-passing framework of Shirani and Bayati [2024], we incorporate higher-order moments of observed outcomes and treatment probabilities to estimate the total treatment effect, without requiring knowledge of the interference network. Our approach leverages machine learning techniques to extract informative patterns from these higher moments, enabling our estimator to capture complex counterfactual behaviors, including non-monotonic trends in outcome means relative to treatment proportions. While we demonstrate strong performance across various outcome specifications and network structures, the framework's applicability may be limited when multiple outcome observations are unavailable.

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

# A   Supplementary Experiments

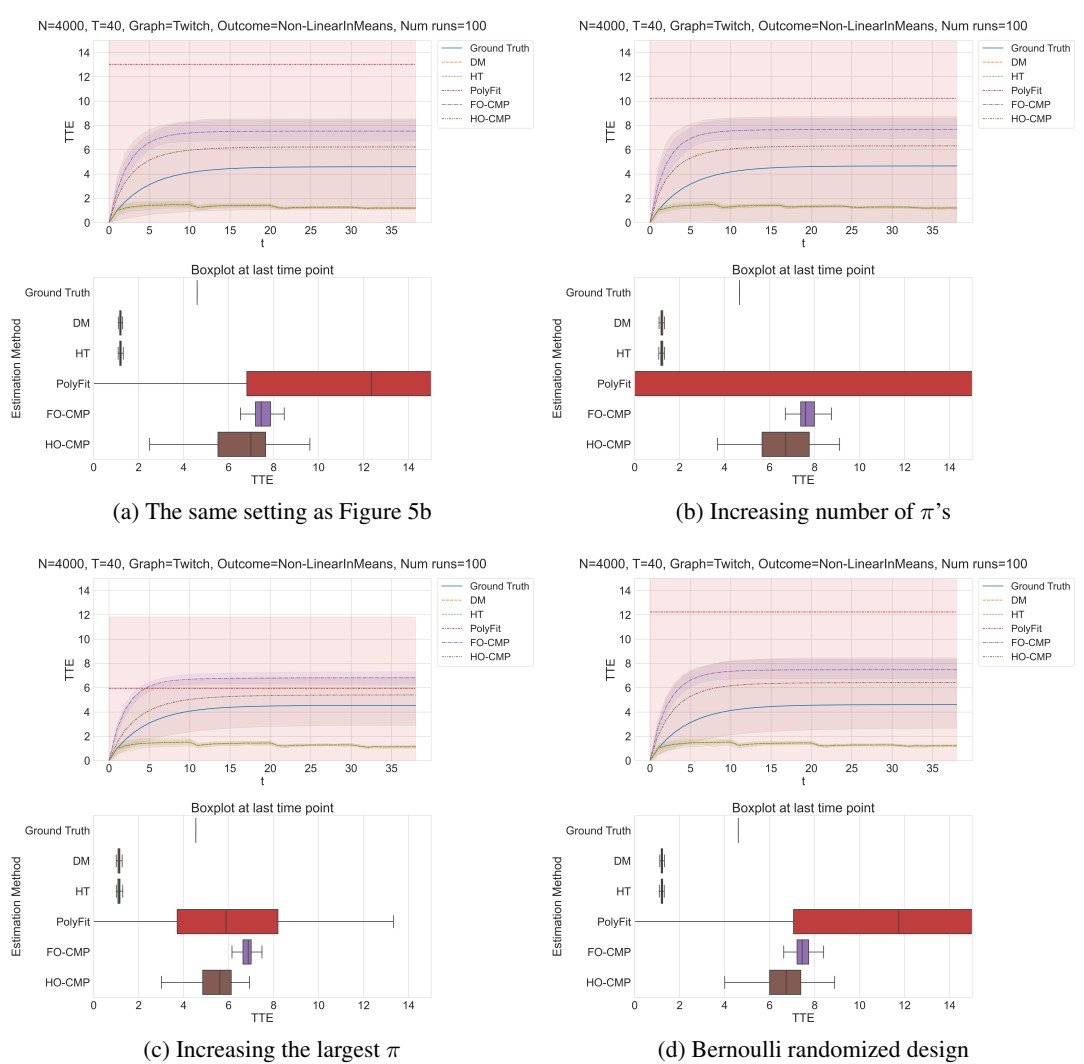

(a) The same setting as Figure 5b

(b) Increasing number of $\pi$'s

(c) Increasing the largest $\pi$

(d) Bernoulli randomized design

Figure 6: *Robustness check, under the Non-LinearInMeans with Twitch graph and $T = 40$, i.e., setting of Figure 5b.* (a): Original Figure 5b. (b): Increasing $L$: i.e., $\pi^{(\ell)} = 0.1\ell$ and $T^{(\ell)} = 8\ell$ for all $\ell \in \{1, \ldots, 5\}$. (c): Increasing treatment probabilities, i.e., $(\pi^{(1)}, \pi^{(2)}, \pi^{(3)}, \pi^{(4)}) = (0.1, 0.2, 0.4, 0.6)$. (d): Using Bernoulli randomized design.

