# OpenReview forum: "Higher-Order Causal Message Passing for Experimentation with Complex Interference"
_NeurIPS.cc/2024/Conference — NeurIPS 2024 poster_

### Official Review · Reviewer_YRHE · 2024-06-29

**Soundness:** 3
**Presentation:** 2
**Contribution:** 2
**Rating:** 3
**Confidence:** 4

**Summary:**

The paper considers the problem of total treatment effect estimation under interference, when the interference structure is unknown.  It proposes an estimation approach based on fitting a data-driven model for low-dimensional dynamics of the mean and variance of observed experimental outcomes over time, motivated by state evolution dynamics.  These state evolution equations come from approximate message-passing (AMP) applied to the potential outcome dynamics chosen in Equation 2.  To produce total treatment effect estimates, initial means and variances are observed from the experiment and propagated forward via the learned state evolution model.

**Strengths:**

- The paper investigates an important unsolved problem in causal inference.
- The data-driven model of the state evolution equations does not rely upon exact functional forms of the dynamics and instead attempts to learn the appropriate dynamics from the observed data.  This increases methodological flexibility by allowing a portion of the dynamics, in addition to the interference, to remain unknown.
- The empirical analysis includes experiments beyond the setting under which the estimator is derived ( e.g. interference occurs across binary edges, the outcome generating process used, and staggered rollout design), increasing the relevance for practice.  Improvements are seen over the few available baselines for the unknown interference setting.

**Weaknesses:**

- The authors should clarify the scope of their algorithm.  Equation 2 defines specific dynamics and does not meet the generality emphasized in the introduction and abstract around equilibrium and interference.  These dynamics do not obviously cover what may be occurring in real-world systems (e.g. lacking memory and being additive).  The authors motivate their paper as relaxing assumptions, but introduce their own alternative assumptions in Section 2.1.  The necessity of those assumptions is also undiscussed, leaving the range of applicability unclear.  For example, the interference matrix is assumed iid Gaussian but the experiments contain binary edges.
- While past research Causal Message Passing is cited prominently in the paper, the paper could use more discussion over what is novel and different here.  To my understanding, the novel contribution is primarily the HO-CMP algorithms introduced in Section 3.  While the experiments demonstrate improvements by using HO-CMP, the idea of data-driven modeling of the state evolution equations appears a relatively straightforward extension.
- The HO-CMP algorithms in Table 1 only leverage simple feature specifications that are low-dimensional.
- Algorithm 1's computational complexity and scalability is unaddressed.
- Overall the paper could use increased clarity.  Many complex ideas are mentioned and occasionally described in insufficient detail.  For example, the authors do not justifying the necessity of considering state evolution instead of modeling Equation 2 directly.

**Questions:**

1.  Why only consider the mean of W_t as a feature in Table 1?
2.  Equation 3: sums should be over i not n?
3.  What are the initial conditions of the counterfactual estimation part of Algorithm 1?
4.  In Equation 2, are the diagonal entries (i=j) treated the same as the off-diagonal entries?  What is the motivation for this?

**Limitations:**

Limitations are adequately addressed.

---

> ### Author Response · Authors · 2024-08-07
>
> We thank the reviewer for carefully reading our paper and providing insightful comments. Below we provide responses to the weaknesses and questions you raised.
>
> Response to Weaknesses:
>
> Thanks for helping us clarify the scope of our algorithm. In the introduction, we highlighted our algorithm can generalize the existing estimation methods under network interference in two aspects. First, our algorithm efficiently uses the data before the system reaches equilibrium. Second, our algorithm does not require knowledge of the network. Indeed Equation (2) encompasses these two aspects. We use data before equilibrium so that there are T periods in total (not just the last period at equilibrium). Moreover, we do not require the knowledge of the interference matrix G in Equation (2).
> In addition, the properties of lacking memory and being additive can be covered in function g_t(). Specifically, lacking memory happens when the coefficient of Y_t^j is zerol and the coefficient of W_0^j, …, W_t^j are zerol. Being additive occurs when g_t() is a linear function of Y_t^j, W_0^j, …, W_t^j.
>
> The Gaussian assumption of the interference matrix is imposed to theoretically prove the state evolution in Equation (4). However, a key point of this paper is to show that the treatment effect estimation based on state evolution is accurate for general graphs, including the cases where the interference matrix does not follow the Gaussian distribution. Therefore, in the experiments, we consider the graphs with binary edges and illustrate the benefit of using our proposed method.
>
> Thanks for helping us be more clear about our contribution as compared to [SB23]. It is true that the main contribution is in Section 3 that provides more general and more accurate estimation methods of treatment effects. Our algorithm is more general in the sense that we allow for any number of \pi’s, whereas [SB23] only accounts for two different \pi’s in the randomized experiment. Our algorithm is more accurate because of two conceptual differences with respect to [SB23]. First, we use the state evolution of both the first and second moments (\nu_t and  \rho_t^2), while [SB23] only uses the state evolution of the first moment. Second, we use a data-driven approach to learn g_t(), while [SB23] considers a linear specification of g_t().
> You are right that the feature specification in Table 1 only include first and second moments. We have explored the inclusion of higher-order moments, but the variance and MSE are generally higher than the three versions of HO-CMP in Table 1.
> Our algorithm runs in time O(NT). Once the mean and variance are calculated for every time t (that requires linear runtime in both N and T) in the step 1, the remaining steps only use the mean and variance. Therefore the algorithm is scalable with N (as in practice N is generally much larger than T).
>
> Thanks for helping us be more clear. The state evolution equations are derived based on Equation (2) that hold for a broad class of unknown functions g_t() and unknown interference matrix G. As state evolution equations only involve summary statistics, they provide potential for developing efficient treatment effect estimation methods (especially, more efficient than [SB23]). This is exactly the problem studied in this paper.
>
> Response to Questions:
>
> W_t^j is binary. The higher-order moments are therefore the same as the mean.
>
> You are right. We will correct it in the revision.
>
> We set the initial conditions (i.e., \nu(0), \nu(1), \rho^2(0), and \rho^2(1) at time 0) as the mean and variance of observed outcomes at time 0 (i.e., \nu(0) and \nu(1) are set as the same; similarly, \rho^2(0) and \rho^2(1) are set as the same). We then compute \nu(0), \nu(1), \rho^2(0), and \rho^2(1) for time t=1, …, T. The difference between \nu(1) and \nu(0) is the estimated TTE at time t. As one can expect, the estimation of TTE is more accurate as t increases.
>
> You are right that the diagonal and off-diagonal entries are treated the same. It is possible to treat them differently and then separately identify the direct and spillover effects. However, as our goal is to identify TTE, it is sufficient to consider the model in Equation (2) and then the estimated effect will be the sum of direct and spillover effects.

---

> > ### Comment · Reviewer_YRHE · 2024-08-09
> >
> > I thank the authors for the clarifications in the rebuttal, but my opinion on the paper overall is unchanged.

---

### Official Review · Reviewer_V4GS · 2024-07-07

**Soundness:** 1
**Presentation:** 2
**Contribution:** 1
**Rating:** 3
**Confidence:** 5

**Summary:**

This paper discusses the causal effect estimation under spatiotemporal interference characterized by a dynamic system. The authors are devoted to improving the causal message-passing framework by adding interaction terms of inputs in the summary function $f_\theta$ (a linear regressor). Synthetic and semi-synthetic experiments are conducted to show the performance of the proposed method.

**Strengths:**

1. The authors explicitly present their methodology with clear definitions and notations.
2. The experiment results are summarized clearly.

**Weaknesses:**

1. The comparison with the most related work [1] is very important but scarce, and I think the marginal contribution is little.
The basic framework is all the same with [1], and the scanty innovation of this paper is the so-called "higher order", which is formulated as two more interaction terms and a square term. I don't think these additional input terms are necessary when given a non-linear model $f_\theta$, especially when the authors don't develop any theory that necessitates the simplicity of linear regression.

2. The literature review is disordered, and there are apparent misunderstandings for certain important works. I list some examples in the following.
- The taxonomy "Restrictions on interference structure" and "Treatment effect dynamics and temporal interference" overlap severely, wherein [2] studies the general temporal interference characterized by a MDP, while the authors list this paper with other two-sided marketplace paper in "Restrictions on interference structure".
- I don't understand why [3] and [4] are discussed in the category of "Partial interference", since these two papers are important and don't belong to this category.

3. More discussion on the basic setting is needed. I think neither the authors nor [1] discuss the concrete and applicable scenario of the dynamic system to show it's indeed meaningful, especially when they are agnostic to the network. I take network interference as an instance for the cross-sectional part. As the number of units increases, how the network topology change indeed impact the scenario, e.g. whether the degree of units increases with $n$ (Erdos-Renyi) or just keeps constant? The details behind the asymptotics indeed matter.



Ref.
[1] Sadegh Shirani and Mohsen Bayati. Causal message passing: A method for experiments with unknown and general network interference. arXiv preprint arXiv:2311.08340, 2023.
[2] Vivek Farias, Andrew Li, Tianyi Peng, and Andrew Zheng. Markovian interference in experiments. Advances in Neural Information Processing Systems, 35:535–549, 2022.
[3] Christina Lee Yu, Edoardo M Airoldi, Christian Borgs, and Jennifer T Chayes. Estimating the total treatment effect in randomized experiments with unknown network structure. Proceedings of the National Academy of Sciences, 119(44):e2208975119, 2022.
[4] Johan Ugander, Brian Karrer, Lars Backstrom, and Jon Kleinberg. Graph cluster randomization: Network exposure to multiple universes. In Proceedings of the 19th ACM SIGKDD international conference on Knowledge discovery and data mining, pages 329–337, 2013.

**Questions:**

I have no questions.

**Limitations:**

No, the limitations of this paper are discussed inadequately, as discussed in weaknesses.

---

> ### Author Response · Authors · 2024-08-07
>
> Thank you for reading our work. As outlined in our global response, this research extends the estimation method of [SB23] by adopting a more data-driven approach. We will carefully revise the manuscript to ensure that the unique contributions of our work are clearly discussed and distinguished.
>
> Regarding the literature review, we first emphasize that all the mentioned works have made significant contributions to the network interference problem. Specifically, [FLPZ22] is an excellent work focusing on experiments in dynamical systems. However, as clarified by the authors, their approach considers systems where treating some units impacts others through a limiting constraint, which is a **special case** of interference structure between units. By "Restrictions on interference structure," we refer to works that address the interference problem within a particular setting. Meanwhile, "Treatment effect dynamics and temporal interference" refers to studies that examine variations in the treatment effect over time. Additionally, [YABC22] and [UKBK13] are both crucial papers that we cited in the section related to "Partial interference," even though they do not adhere to the partial interference assumption. Indeed, these works effectively highlight the challenges associated with the broad applicability of studies in the "Partial interference" category and propose innovative solutions to mitigate these issues. We will ensure that the presentation is clearer to avoid these types of confusion.
>
> About the core of the theoretical foundations of Causal-MP which proves, under certain assumptions, as N grows, sufficient statistics (means and variances) of the outcomes evolve according to state evolution equations. We also provide empirical simulations under network structures and outcome specifications that are studied in the prior literature with diverse network topology (e.g. Linear-in-means on random graphs studied by Leung and we generalize it to non-linear specifications and on real network data). For additional details please see Section 5 of [SB23].

---

### Official Review · Reviewer_YfYx · 2024-07-08

**Soundness:** 3
**Presentation:** 4
**Contribution:** 1
**Rating:** 3
**Confidence:** 3

**Summary:**

This paper studies experimental interference, in which the treatment assignments of one unit affects the outcome of another. The majority of work in this area assumes interference acts through a network, and requires knowledge of that network to reduce bias in the resulting estimator. This paper takes a general approach in which interference occurs due to an unknown graph, and a unit's outcome can be affected by its neighbors' treatment status as well as their past outcomes. With such a flexible potential outcomes model, it is very difficult to estimate the total treatment effect in an unbiased way. The paper suggests an extension of the causal message passing algorithm that estimates the state evolution equations of the potential outcomes using higher-order functions of the past treatment status and outcomes, as opposed to the original CMP paper which used linear models of these quantities.

**Strengths:**

The paper provides thorough experimental validation of the proposed method. In particular, the method is evaluated on simulated and real graphs, and is compared against three baselines including Cortez et al's polynomial fit method. The experiments are simulated under the staggered rollout design, which is of interest in the academic literature as well as in practical industry settings.

The ideas in the paper are presented well, with a high quality of writing throughout.

**Weaknesses:**

If I understand correctly, this work extends the CMP algorithm of Shirani and Bayati by using nonlinear estimators of the state dynamics. It is unclear to me that this delta is a significant enough improvement over the original CMP method to merit publication at Neurips.

I reviewed the Shirani and Bayati paper, and am concerned at how closely the writing, structure, and in some cases individual sentences of this paper parallel that work. I will leave it to the AC to make any recommendations about copyright / academic integrity, but here are the passages of most concern:
1. The phrase "Inspired by the literature on AMP algorithms, we refer to (4) as state evolution equations of the experiment" appears verbatim in the original CMP paper.
2. The related works of this paper appears to be a reworded and edited version of a subset of the related works of the CMP paper. In particular, the paragraphs beinning on lines 124 and 134 almost exactly mirror their counterparts in the CMP paper.

**Questions:**

1. Is the HO-CMP-I algorithm the same as that of Shirani and Bayati? If not, I would suggest including the original CMP algorithm as a baseline.
2. [Authors do not need to provide new experiments; discussion would suffice.] The HO-CMP algorithm outperforms polyfit as shown in Figure 1. It's unclear to me if this is due to increased regularization (maybe a 2-degree polynomial provides a better approximation to a half-sine than a 4-degree polynomial on the data in Figure 1) or an inherent property of the HO-CMP algorithm (maybe the claim is that not all types of interference are captured by the polyfit algorithm?). I would be curious if the authors had some intuition here, perhaps an example where polyfit wasn't obviously overfitting, but HO-CMP still outperformed polyfit.

**Limitations:**

Yes

---

> ### Author Response · Authors · 2024-08-07
>
> Thank you for your thoughtful comments. Indeed, the reviewer is right, and as explained in our global response, this work builds on the foundation laid by [SB23] by extending their estimation method. We would also like to clarify that there was no intention to mirror the writing of [SB23]. However, it was necessary to discuss the problem setup and relevant literature. In the revision, we will ensure that the differences in the writing are clear and that the unique contributions of this work are highlighted.
>
> To compare HO-CMP and PolyFit, it is important to note that HO-CMP utilizes more data by incorporating the evolution of the experiment over time. In contrast, PolyFit relies solely on the outcome observation at the equilibrium point where the treatment effect stabilizes. As a result of this additional data, HO-CMP can effectively mitigate the issue of overfitting that often affects PolyFit. Additionally, HO-CMP-i can be considered as a generalization of [SB23] estimator, allowing for more than two stages in the experiment.

---

> > ### Comment · Reviewer_YfYx · 2024-08-09
> >
> > Thank you for your response.

---

### Official Review · Reviewer_p2A8 · 2024-07-12

**Soundness:** 3
**Presentation:** 3
**Contribution:** 3
**Rating:** 5
**Confidence:** 4

**Summary:**

The authors motivate a regression framework using Causal Message Passing to estimate global average treatment effects under some unknown forms of interference. The simplest way to think about their proposed algorithm is that CMP is used to motivate particular sufficient statistics for the interference dynamics which are fed into a regression model to perform corrections and enable estimation.

**Strengths:**

The CMP framework is, I think, a very interesting direction for research. In particular, it's a very important direction to weaken the kinds of assumptions about interference necessary to make when working with experiments.
I also think there's something convenient in working with high-level statistics from experiments (for example, many large companies' experimental systems makes these particularly easy to retrieve, so reflecting that makes it easier to integrate proposed methods).

**Weaknesses:**

The biggest problem with this paper is that it doesn't actually set out theoretical scope conditions for when it works. I'll be a bit more specific on two points:
1. The combination of the definition of TTE_t as the large sample limit combined with HO-CMP-I under Bernoulli randomization. Suppose W_t is determined by Bernoulli randomization (with a fixed probability, \pi across all time periods). Then as N \to \infty, then \bar{w}_t \to \pi. This means that the relationship between \bar{w}_t and \hat{\nu}_t will always be exactly zero. Even under the useful linear form discussed on line 228, it appears that this method could not possibly learn anything effectively. I suspect, in fact, that this would also be problematic in finite sample sizes (see below notes on evaluation). Critically, however, you haven't spelled out any of these scope conditions, so the paper does not make clear what is necessary to achieve success. And I will reiterate that this is using the very asymptotics gestured to in the paper and an experimental design which is explicitly discussed.
2. More specifically, I will return to line 228, which gestures at a generative model under which the proposed method (HO-CMP-I) should work well. But what does this mean? Should we expect the estimator to be unbiased? Should we expected it to be BLUE? Should we expect it to be semi-parametrically efficient? It would be ideal to have theoretical results around this, but barring that, these questions should be addressed directly and explicitly in the experiments.

On the subject of the experiments, I want to see more specifics than the charts you've provided give. Are the estimators unbiased? What are their MSEs? How do these evolve as a function of sample size? What are their convergence rates? In the absense of theoretical results, it is critical to have clear answers to these kinds of questions.

Some minor points that I don't think matter much at all, but will share regardless:
- I think "higher-order" is an overly confusing way to define this approach. When I think higher-order, I think about something like a Taylor series: using more granular information about the setting to define a better estimator. In contrast, your approach is doing something more akin to higher-_aggregation_. I would consider using a term like "higher-level" or something which connotes this kind of aggregation.
- On line 211 you refer to "multi-label" ML models, but I think you mean "multi-task" models or "multivariate" models.
- What you've done here doesn't seem obviously like "machine learning." While I have no problem with linear regression, what it appears that you've done here
- line 264: you don't need to tell the reader what \pi is :)

**Questions:**

see above

**Limitations:**

see above

---

> ### Author Response · Authors · 2024-08-07
>
> 1. We thank Reviewer p2A8 for helping us make this point more clear. The framework of HO-CMP enables us to learn an arbitrary functional form of \nu_{t+1}(w). In HO-CMP-i, we specifically model this as a linear function of \bar{w}_{t+1}. It is important to note that \pi varies during the experiments. For example, in the setting of Figure 2, we set (\pi^{(1)}, \pi^{(2)}, \pi^{(3)}, \pi^{(4)})=(0.1,0.2,0.4,0.5) for four intervals of duration 10, each. This approach allows us to learn the coefficient of \bar{w}_{t+1} in the linear function.
>
> 2. As we explained in the global response, the primary focus of this paper is to develop estimators and validate their performance in practical applications rather than provide theoretical analysis. The reason is that the theoretical treatment of valid questions (such as the ones raised by the reviewer) are highly challenging in this setting. We need to adjust our expectations as in the setting studied here (involving pervasive interference and unknown network structures) it is very difficult to estimate TTE. For example, take the Non-LinearInMeans outcome specification that we introduce as a generalization of the LinearInMeans (studied in the literature, e.g., Leung 2022). To the best of our knowledge, the proposed HO-CMP framework is the first method to efficiently utilize data for estimating TTE in this setting, and PolyFit and the estimator of [SB23] which is similar to HO-CMP-i being also the best benchmarks one could find (no other estimator to our knowledge can be applied here).
>
> 3. We thank Reviewer p2A8 for the comments. The expected convergence rate is O(1/\sqrt{N}), as discussed in [SB23]. We agree that an empirical demonstration of the rate would be a valuable addition to the paper that we plan to include in the revision.
> We agree with the reviewer that the term “higher-order” might be confusing. It is intended to highlight that our function approximation of the dynamics incorporates additional terms. We will address all minor comments to clarify any potential ambiguities.

---

> > ### Comment · Reviewer_p2A8 · 2024-08-11
> >
> > I thank the authors for their clarifications. I will retain my score.

---

### Author Response · Authors · 2024-08-07
**Global comment**

We thank the review team for reading our work. Here, we outline our responses by clarifying the contribution of the current work and defer more details to the reviewers' responses. This work builds on the foundation established by [SB23], extending their method in two directions by introducing the family of Higher-Order Causal Message Passing (HO-CMP) algorithms.

First, HO-CMP incorporates higher-order moments of unit outcomes, unlike [SB23]'s approach, which only employs the first moments for estimation. Second, while [SB23] focus solely on two-stage experiments, our work leverages the additional data provided by having more than two experimental stages, which aligns with the common practice in the tech industry of rolling out treatments through a sequence of experiments [KTX20].

Overall, this paper emphasizes a data-driven approach to the causal message passing framework and the efficient use of experimental data. Additionally, we introduce the Non-LinearInMeans outcome specification, where the spillover effect is non-monotone in the fraction of treated neighbors; as an example of a complex treatment effect structure, we demonstrate how HO-CMP successfully estimates the total treatment effect by effectively utilizing higher-order moments of unit outcomes.

[SB23] S. Shirani and M. Bayati, Causal Message Passing: A Method for Experiments with Unknown and General Network Interference

[KTX20] Kohavi, R., Tang, D., and Xu, Y. (2020). Trustworthy online controlled experiments: A practical guide to a/b testing. Cambridge University Press.

---

### Decision · Program_Chairs · 2024-09-25

**Decision:**

Accept (poster)

**Comment:**

This work mirrors somewhat closely the work of Sadegh Shirani and Mohsen Bayati (https://arxiv.org/abs/2311.08340) and in light of that (beside concerns of plagiarism or self-plagiarism, since I don't see the authors' identities), the contribution of this submission remains a bit incremental.